# Differential HIV-1 Proviral Defects in Children vs. Adults on Antiretroviral Therapy

**DOI:** 10.3390/v17070961

**Published:** 2025-07-09

**Authors:** Jenna M. Hasson, Mary Grace Katusiime, Adam A. Capoferri, Michael J. Bale, Brian T. Luke, Wei Shao, Mark F. Cotton, Gert van Zyl, Sean C. Patro, Mary F. Kearney

**Affiliations:** 1HIV Dynamics and Replication Program, Center for Cancer Research, National Cancer Institute, Frederick, MD 21702, USA; mkatusii@fredhutch.org (M.G.K.); adam.capoferri@nih.gov (A.A.C.); mib4004@med.cornell.edu (M.J.B.); 2Frederick National Laboratory for Cancer Research, Leidos Biomedical Research, Inc., Frederick, MD 21702, USA; lukebria@mail.nih.gov (B.T.L.); sean.patro@nih.gov (S.C.P.); 3Division of Medical Virology, Department of Pathology, Stellenbosch University, Cape Town 7500, South Africa; mcot@sun.ac.za (M.F.C.); guvz@sun.ac.za (G.v.Z.)

**Keywords:** HIV in children, HIV reservoir, HIV persistence, HIV proviruses, HIV latency, intact provirus

## Abstract

HIV-1 proviral landscapes were investigated using near-full-length HIV single-genome sequencing on blood samples from five children with vertically acquired infection and on ART for ~7–9 years. Proviral structures were compared to published datasets in children prior to ART, children on short-term ART, and adults on ART. We found a strong selection for large internal proviral deletions in children, especially deletions of the *env* gene. Only 2.5% of the proviruses were sequence-intact, lower than in the comparative datasets from adults. Of the proviruses that retained the *env* gene, >80% contained two or more defects, most commonly stop codons and/or *gag* start mutations. Significantly fewer defects in the major splice donor site (MSD) and packaging signal were found in the children on short or long-term ART compared to the adults, and *tat* was more frequently defective in children. These results suggest that different selection pressures may shape the proviral landscape in children compared to adults and reveal potentially different genetic regions to target for measuring the intact HIV reservoir and for achieving HIV remission in children.

## 1. Introduction

Human immunodeficiency virus type-1 (HIV-1) in infants and children remains a significant global health challenge despite effective antiretroviral therapy (ART) for prevention and treatment. In 2023, 16% of pregnant women living with HIV-1 had no access to ART [1,2], and this number could be rising today. Currently, approximately 1.5 million children aged 0–14 years are living with HIV-1, of which only 57% had access to ART in infancy [1]. Without ART, 50% of children born with HIV-1 typically die by 2 years of age [3].

As there is currently no scalable cure, life-long ART adherence is required for children with HIV-1. In adults living with HIV-1, non-AIDS-related morbidities and drug toxicities have been observed in those on ART [4,5,6]. However, little is known about the long-term effects of HIV-1 and ART in children. HIV-1 reservoirs (the intact proviruses that persist on ART) in children with perinatally acquired infection are influenced by maternal ART and immune status during pregnancy, early-life immune dynamics, and timing of ART initiation [7,8,9,10]. Current treatment guidelines recommend that infants perinatally exposed to HIV-1 should begin combination ART within 6 h of birth [11,12]. Very early ART initiation limits the size of the viral reservoir, as reservoir seeding occurs during uncontrolled viral replication [13,14,15,16,17]. Infants who initiate ART before 6 weeks of age have smaller proviral reservoirs than those treated later [17,18,19,20,21]. Proviral DNA levels decay following ART initiation until reaching a stable set point [19,22,23,24,25,26,27,28,29], although a new case study suggests that reservoir levels may decay further after decades of ART in some individuals born with the virus [20].

Previous studies have investigated HIV-1 dynamics in a subset of children participating in the Children with HIV Early antiRetroviral (CHER; NCT00102960) trial [30]. ART was initiated at a median age of 5 months and viremia was suppressed for a median of 8 years at sampling. Van Zyl et al. found no evidence of viral replication while on ART in the children [31] and Katusiime et al. reported that only about 1% of persisting proviruses on long-term ART were genetically intact [32], a lower proportion than in most adults on long-term ART [33,34,35]. Integration site analyses demonstrated that infected cell clones were present in this cohort as early as 1.8 months of age and persisted for at least 9 years on ART [36]. It was also demonstrated that the largest infected cell clones were stable over time and contained solo HIV LTRs with no internal HIV sequences [37]. Additional pediatric studies have described proviral structures in pre-ART and during short-term ART and found that intact proviruses declined from ~50% to ~20% over the early months of treatment [38]. In addition, ART initiation prior to one week of age was associated with lower reservoir sizes [26].

Our current study addresses an important gap in the field, characterizing the structures of proviruses with internal HIV sequences that persist in children who adhere to long-term ART. Although previous studies have shown that the largest infected cell clones have no internal HIV sequences (i.e., only solo LTRs), comprising about 50–90% of the infected PBMCs in children on long-term ART, and that only about 1% of the infected cells contain inferred sequence-intact proviruses, the structures of proviruses in the remaining infected cells remains unknown. Characterizing the proviruses that contain HIV-1 gene-coding sequences on long-term ART may inform of the potential selection pressures that eliminate some infected cells while allowing others to persist, ultimately revealing new targets for eliminating the HIV-1 reservoir in children. In-depth characterization of the HIV-1 reservoir, as well as the defective proviral population that persists on ART, is vital to identifying the sources of the viral reservoir and to inform future curative strategies in children. Here, we investigated the genetic structures of proviruses persisting in children on long-term ART and compare them to proviral structures present in children prior to ART initiation, on short-term ART, and to those that persist in adults [33,34,35,38].

## 2. Results

### 2.1. Fewer Proviruses in Children on Long-Term ART Are Sequence-Intact Compared to Children on Short-Term ART

PBMCs were obtained from five children living with HIV-1 and on long-term ART (7.4–8.8 years) who were enrolled in the CHER trial (NCT00102960) (Table 1). A total of 882 (range 56–450) near-full-length (NFL) proviral sequences were obtained from the five children in the CHER cohort who had viremia sustainably suppressed on ART for a median of 8 years (Table 2) (GenBank Accession: PV625462-PV626342). Of these sequences, 860 were defective and 22 (2.5%) were inferred intact using the ProSeq-IT multi-parameter intactness tool [39]. We compared the 882 proviral structures in children on long-term ART (7.4–8.8 years) to previously published NFL proviral sequence datasets obtained from children with varying times of ART initiation and durations of treatment (Table 3). These comparative datasets included proviruses in infants prior to ART initiation (n = 132 sequences) (EIT study, pre-ART) [38], in children on short-term ART (<1 year) after initiating treatment < 5 days after birth (n = 244 sequences) (EIT study, on-ART) [38], and in children on short-term ART (<2.5 years) after receiving delayed treatment (<1 year old) (n = 132 sequences) (EIT control group, delayed ART) [38]. We also compared the proviral structures from children on long-term ART to proviral structures in adults on ART for a median of 9.26 years (n = 1056 sequences) [33,34,35].

We found that children on long-term ART have significantly fewer sequence-intact proviruses than children on short-term ART who received either early or delayed treatment (2.5% vs. 19.6% and 6.1%, *p* < 0.0001, Fisher test) and have fewer intact proviruses than the adults on ART analyzed with a similar method (2.5% vs. 5.9%) (*p* = 0.0002, Fisher test) (Table 4). Of the defective proviruses in the CHER children (long-term ART), 723 (82%) contained large internal deletions (proviruses < 7 kb), slightly higher than those reported for adults on ART (71%) (*p* = 0.046, Fisher test), and greater than the fraction with large internal deletions in children on short-term ART (46–61%) (*p* < 0.0001, chi-squared test) or pre-ART (19%, Fisher test) (*p* < 0.0001) (Figure 1). Of the pre-ART proviruses obtained from infants in the EIT study, the majority (55%) were inferred intact. Defective proviruses resulting from mutations or small deletions comprised 16% of sequences in the CHER cohort (children on long-term ART) compared to 34% in children on short-term ART (*p* < 0.0001, Fisher test) and 23% in adults on ART (*p* = 0.0012, Fisher Test). This finding suggests a strong selection with time against proviruses in children that are >7 kb in length, regardless of whether they are intact or defective.

### 2.2. Partial or Complete Env Deletions Were Found in Most Defective Proviruses Regardless of ART Status or Duration

Further investigation of the defective sequences with large internal deletions (proviruses < 7 kb) revealed that the majority contained complete or partial deletions of the *env* gene in all datasets, regardless of ART status or duration (Figure 2). Proviruses with completely deleted *env* genes comprised 40–56% of all sequences < 7 kb. Large internal deletions that spanned both the 5′ and 3′ halves of the genome, including partial/complete deletions of *env*, comprised 32–52% of proviruses < 7 kb, while deletions in only the 5′ half of the genome were seen in 3–12% of proviruses < 7 kb (*p* < 0.0001, binomial test). These data demonstrate that there is a strong selection pressure against the maintenance of the *env* gene and that this selection pressure exists prior to ART and during short-term and long-term ART. Of the proviruses with large internal deletions that persist on ART, about 90% (blue + yellow; Figure 2) include at least a partial deletion of the *env* gene, suggesting *env* confers a deleterious effect on infected cell clone survival, independent of the intactness of the proviral genome.

### 2.3. Most Proviruses > 7 kb in Children on Long-Term ART Contain Multiple Fatal Defects

All proviruses that were determined to be defective using the ProSeq-IT tool but did not contain large internal deletions (n = 137 sequences) were analyzed for their number of defects (Figure 3). While most of the proviruses in this subcategory collected prior to ART contained only one defect (51%), most proviruses in the long-term, on-ART datasets contained two or more defects (>81%) (EIT pre-ART vs. CHER *p* < 0.0001 or vs. Adults *p* < 0.0001, Binomial test). The highest variance in observed defects was seen in the EIT on-ART dataset, where some proviruses had up to nine defects. These data show that proviruses with single defects are selected against with time on ART, leaving persistent proviruses with multiple defects in both children and adults on long-term treatment.

### 2.4. Different Defects in Proviruses That Are >7 kb in Children vs. Adults on ART

Of the 137 defective proviruses that were >7 kb in the CHER cohort, frameshift-induced stop codons, stop codons in the absence of frameshifts, and *gag* start codon mutations were the most common defects, consistent with the defective proviruses in all infant and child cohorts (Figure 4). However, while mutations in the major splice donor site (MSD, red) and packaging signal (Psi, light orange) defects were very common in adults on ART (~38%), they were less frequent in children before and on both short- and long-term ART (<21%) (*p* < 0.0001, binomial test). The lower frequency of Psi defects in children on ART may influence results of the Intact Proviral DNA assay (IPDA) [40] by having a higher percentage of proviruses that are Psi+. The IPDA is a high-throughput method used to estimate the number of intact proviruses in a sample using the presence of Psi and Rev response element (RRE) sequences as a surrogate for NFL sequencing, since Psi and RRE are commonly deleted in proviruses that persist on ART. Although a higher fraction of defective proviruses >7 kb in children on ART retain the packaging signal compared to adults on ART, most proviruses contain large internal deletions, including deletions of RRE, preventing these proviruses from scoring positive by IPDA. Another observed difference between persistent proviruses in children on ART compared to adults on ART is the higher frequency of insertions in children that render the proviruses defective (2.5% vs. 0.7%) (*p* = 0.021, binomial test). In total, we observed the accumulation and persistence of proviruses with multiple defects with time on ART and we observed varying frequencies regarding types of defects in children compared to adults, although larger studies are needed to confirm these findings.

### 2.5. Defects of Tat Are Observed in Proviruses That Retain Env in Children

*Tat*/*rev* coding regions have overlap with the *env* gene in the 3′ region of the HIV genome. We analyzed defective proviruses that retained *env* for possible *tat* and *rev* defects (Figure 5). In the EIT pre-ART cohort, most “*env*-retaining” proviruses contained inferred-intact *tat* genes (71%), while in the EIT on-ART cohort, defects in *tat* were observed in 65% of the *env*-retaining proviruses (*p* < 0.0001, Binomial test) (Figure 5A). Interestingly, the loss of *tat* did not correlate with time on ART, with children on long-term ART (CHER) having only 40% of *env*-retaining sequences being defective in *tat*. This difference may reflect the large number of proviruses that delete the *env* gene entirely and therefore render *tat* defective as well. Surprisingly, we found no correlation between ART treatment and defective *rev* genomic regions in the *env*-containing proviruses. Instead, we found that *rev* was retained in 50% of *env*-containing proviruses in children both before and during ART (Figure 5B). Retention of *rev* in on-ART cohorts may allow for vRNA/protein production, which should be a target of future studies towards achieving HIV-1 remission without ART. Unique visibility of such epitopes, even on ART, may present an opportunity for immunotherapy to boost reservoir elimination.

## 3. Discussion

Understanding the impact of the timing of ART initiation and the duration of suppressive ART on persistent provirus genetics is important for defining targets for curative strategies, especially in children with perinatally acquired HIV-1 who face a lifetime on ART. Previous studies have characterized the proviral reservoirs of infants and children; however, the defective proviral landscape on long-term ART was not fully profiled [32,33,34,35,38]. In this study, we performed a deep characterization of the proviral landscape in children in the CHER cohort (treated within 2.7–10.9 months of age) and on long-term ART (84–108 months), compared to previously published datasets of children with differing times of ART initiation and duration: early-treated infants (EIT) (treated within 0.2 h–4.77 days) on short-term ART (1–24 months) and infants who initiated ART at similar ages to those in the CHER cohort (treated within 2.6–11.7 months) but on short-term ART (16–31 months) [38]. We also compared proviral landscapes in the children in the CHER cohort to those in adults on ART (10–212 months) [33,34,35].

In the samples from the CHER cohort (long-term ART), the majority of proviruses were defective, and only 2.5% were found to be intact. This finding is consistent with previous data from children [32] but is lower than the fraction of intact proviruses reported for adults on ART using a similar approach (6%) [33,34,35]. In contrast, in the “EIT pre-ART” dataset, most proviruses were inferred intact, as is expected in uncontrolled infection. In the on-ART datasets, many-to-most defective proviruses contained large internal deletions regardless of the time of ART initiation or the duration of ART. Of the on-ART datasets, the “EIT on-ART” group (short-term ART) had the smallest fraction of sequences with large internal deletions, consistent with the shortest duration on ART. This fraction was higher in the CHER children with longer durations on ART, suggesting a selection against proviruses that retain internal genes, as previously seen in adults [41,42]. In all datasets, large internal deletions with partial or complete deletion of 3′ genes were most frequently observed, potentially resulting from template switching during reverse transcription. *Env*-deleted genomes have also been commonly observed in studies investigating the proviral landscapes in adults [34,35,43,44], suggesting that *env* expression is highly selected against in the persistence of infected cells on ART in both children and adults [45] and warrants further investigation in future studies. A recent study by Botha et al. found that the largest infected cell clones in children on long-term ART contain only solo LTRs [37], suggesting that the decline of intact proviruses may be due, in large part, to homologous recombination of the LTRs post-integration. This observation aligns with the mechanism of solo LTR formation of human endogenous retroviruses (HERVs), as solo LTRs comprise as much as 90% of the HERVs arising from homologous recombination of endogenous LTRs [46,47,48,49]. Although the NFL single-genome sequencing approach used here does not capture solo LTR proviruses, combined with the previous study by Botha et al., which was performed on the same cohort of children, our results suggest that >90% of proviruses that persist in children for 7–9 years on ART contain large internal deletions of either the *env* gene or of all internal HIV-1 genes.

In the proviruses retaining the *env* gene, the *rev* coding region was inferred to be functionally intact in about 50% in both children and adults, likely due to the flexibility of several structural features of Rev [50,51,52,53,54]. This finding is consistent with a study from Imamichi et al., which reported that some defective proviruses can produce viral proteins that elicit CTL responses, suggesting that HIV unspliced and partially spliced products from defective proviruses are exported from the nucleus in a Rev-dependent manner [50]. Unlike *rev*, *tat* defects appeared to be selected during ART and potentially more so in children than in adults [55]. However, it is important to consider that >85% of all proviruses that persist on ART have large internal deletions that render both *tat* and *rev* defective (not including the solo LTRs). As *tat* and *rev* function in viral expression and *env* encodes a surface protein, these data drive the hypothesis that the ability to express genes in the 3′ region may influence the selection of proviruses by immune pressures, such as CTL and/or neutralizing antibodies, but future studies are required since we cannot make a conclusion due to the small size of the study and the inherent diversity across the cohorts [45,56,57,58,59,60,61,62,63]. The splice acceptor sites required for both the completely spliced *tat*/*rev* and the partially spliced *vpu*/*env* viral RNA species are located in overlapping regions (A4cab/A5 and A7), and, since the sequences of the *vpr*-*tat*/*rev* exon 1 and the *env*-*tat*/*rev* exon 2 containing major splicing elements (such as donor and acceptor sites, enhancers, and suppressors for introns/exons) are relatively close to the *env* gene, a provirus retaining *env* might also maintain *tat*/*rev* expression and downstream splicing. Consequently, these gene products may be good targets for engineered antibodies or T cells designed to achieve HIV-1 remission without ART [64,65]; this may be possible in combination with latency-reversing agents.

The “EIT delayed-ART” group, the CHER children, and the adults all had high frequencies of defects in the *gag* start codon, suggesting that the failure to express *gag* proteins may be favorable for persistence of infected cells on ART. However, the children had lower frequencies of Psi deletions compared to adults. It is not clear whether this difference is related to the differential immune systems of children vs. adults, to the different HIV-1 subtypes in the cohorts studied (HIV-1 subtype C in the children vs. subtype B in the adults), or one of many other possible factors—or whether the differences observed are simply due to the small sample size of the study. However, the variances observed in this study drive the hypothesis that differential selection pressures in children born with HIV vs. adults who acquire HIV may influence the proviral structures and the infected cells that persist on ART. Future larger studies are required to test this hypothesis.

IPDA, and similar assays, is a useful tool to differentiate intact proviruses from defective proviruses using primers and probes that target the Psi and RRE regions of the HIV-1 provirus. A positive IPDA signal indicates an intact provirus [40,66]. Although larger studies are needed, the more frequent retention of Psi in defective proviruses in children on ART suggests that this target may be less appropriate for measuring intact proviruses in children with subtype C infection. Measuring the HIV-1 reservoir in children with subtype C infection may more appropriately target defects in the *gag* start codon, which appear to be more frequent than defects in Psi. Although RRE was not frequently defective in the proviruses not containing 3′ deletions, it is important to note that >85% of the total proviruses in children on ART contained 3′ deletions, including RRE, making RRE an appropriate IPDA target for children, as it is in adults. Based on these results, alternative screening methods in which probes targeting multiple targets, such as Q4PCR [67] or Rainbow proviral HIV-1 DNA dPCR [68], may be more informative.

A comparison of the pipelines used in proviral sequence analysis reported bias in the characterization of defective sequences [69]. As there is no consensus pipeline to classify defective sequences, variance in reported defects leads to an imprecise record of the proviral landscape of people on ART. Furthermore, some pipelines use an order of elimination, in which sequences are binned into defective categories and excluded from further investigation, leading to an underestimation of the frequencies of some defects and an overestimation of others [69]. Here, we used the ProSeq-IT tool, which identifies all known defects in proviral sequence datasets [39]. Using this approach, we found the accumulation of multiple defects in proviruses that persist in both children and adults on ART, with some proviruses containing up to nine different lethal mutations. This observation may be due to limited expression of proviruses with multiple mutations, while some single mutations may be tolerated and permit the expression and detection of viral RNA and proteins. The accumulation of multiple defects may reflect selection against transcriptional/protein production of any HIV genes, thus leaving highly defective, transcriptionally mute proviruses.

Although more than 50% of the proviruses in the pre-ART dataset were inferred intact, stop codons, likely from APOBEC3G/F-derived hypermutation, were the most frequent defect, suggesting that APOBEC3G/F may contribute to pre-ART control of replication in children. On ART, children treated in infancy contained the highest frequency of proviruses with multiple fatal defects, as well as the most variance in defect type. This finding suggests that the proviral landscape undergoes the most change early after ART initiation, as infants experience a rapid decline of HIV-1 DNA due to the culling of infected cells expressing viral proteins. This decline in infected cells results in a shift in the proviral population to those with a wide variety of defects [70], consistent with Koofhethile et al.’s observation of a rapid decline in intact proviruses within one month of treatment initiation [27]. These findings suggest that the time of ART initiation may play a significant role in shaping the proviral landscape, with larger studies needed to verify these dynamics. Guidelines are in place to rapidly treat infants born to mothers with HIV-1; however, adults living with HIV-1 may experience longer durations between initial infection and ART initiation. This difference in the duration of uncontrolled infection prior to ART initiation may play a key role in the differences we observed between the proviral landscapes of children vs. adults, in addition to the fact that the mature immune system of adults may differentially influence the evolution of the reservoir compared to the naivety of the infant immune system.

While this study adds to the limited data on children living with HIV, low sample availability and low proviral loads are a major limitation to generating sequence data. More donors/participants are required. The method of amplicon generation also influences the characterization of the proviral landscape, as recent comparisons of amplification methodologies demonstrated a bias for different sized amplicons [71]. Additionally, our NFL amplification approach precludes the detection of solo LTRs. New methods are needed to capture both solo LTRs and proviruses containing internal genetic information to fully characterize proviral landscapes on ART. Nonetheless, we highlight important differences in the children vs. adult proviral landscape, especially in the more frequent retention of intact MSD and packaging signal in children on short- and long-term ART, and the apparent stronger selection against *tat* retention in children on short-term ART. These findings highlight potential prime targets for the future design of high-throughput assays to measure HIV reservoir size in children and may present opportunities for new strategies to control HIV infection unique to pediatric cohorts.

## 4. Materials and Methods

### 4.1. Donor Samples

PBMCs were obtained from 5 children living with HIV-1 and on long-term ART (7.4–8.8 years) who were enrolled in the CHER trial (NCT00102960, Stellenbosch University HREC M04/07/033A). Additional demographic and virologic characteristics of these children have been reported in van Zyl et al. [31], Katusiime et al. [32], and Bale et al. [36]. The children had their viremia fully suppressed on ART for the duration of treatment, had no evidence of ongoing cycles of viral replication, and contained large, infected cell clones with solo HIV LTRs. The legal guardians provided informed consent for PBMC collection and for the sequencing of HIV proviruses and integration sites. This study was approved by the internal review board at Stellenbosch University (HREC N18/02/020).

### 4.2. Genomic DNA Extraction

Genomic DNA (gDNA) was extracted from 1.25 × 10^6^ PBMCs from each child. In brief, cell pellets of 1.25 × 10^6^ PBMC were subjected to guanidine HCl (Sigma-Aldrich: Rockville, MD, USA, Cat#G9284) lysis, cleaned by isopropanol/ethanol precipitation, and resuspended in 5 mM Tris-HCl (pH 8.0), as described in Katusiime et al. [32]. The resuspended DNA was incubated at 42 °C for 2 h and then stored at −80 °C prior to undergoing near-full-length HIV PCR amplification and sequencing, as described below.

### 4.3. Near-Full-Length HIV Single-Genome Sequencing (NFL-SGS)

Extracted gDNA was diluted to an endpoint (0–1 proviral genomes per well, approximately 10 HIV-positive wells per 96-well plate) with 5 mM Tris-HCl (pH 8.0). Near-full-length (NFL) PCR was performed on each well and was followed by nested PCR (Appendix A). Forward PCR primers were designed in 5′ LTR_U5 and *gag* leader and reverse primers were designed in U5 of the 3′ LTR (Appendix A). NFL amplification was initially performed using Ranger mix (Bioline: Swedesboro, NJ, USA, Cat#BIO-25052) for the material from donors ZA-004, ZA-006, and ZA-011. A 200 μL volume of diluted gDNA was added to a master mix comprising 500 μL 2× Ranger Mix, 284 μL nuclease-free water, and 8 μL each of 50 μM forward and reverse primer. The total volume was spread across a 96-well plate (10 μL per well). PCR cycling conditions are described in Appendix A. NFL amplification from donors ZA-001 and ZA-012 was performed using Platinum SuperFi II DNA polymerase (Thermofisher Scientific: Waltham, MA, USA, Cat#12361010). A master mix containing 210 μL 5× SuperFi buffer, 20 μL of 10 μM dNTP mix, 20 μL each of 10 μM forward and reverse primers, 559.5 μL nuclease-free water, and 10.5 μL SuperFi II DNA polymerase was prepared. A total of 220 μL of diluted gDNA was spread across a 96-well plate (10 μL per well) along with the master mix and was PCR-amplified as described in Appendix A. Invitrogen 96-well 1% agarose E-Gels (Thermofisher Scientific: Cat#G700801) were used to identify positive wells. Amplicons were then run on 12-well 0.8% agarose E-Gels (Thermofisher Scientific: Cat#G501808) along with a 1 kb plus DNA ladder (Thermofisher Scientific: Cat#10488090) to determine amplicon size.

### 4.4. Illumina Sequencing

The resulting amplicons were sequenced by next-generation sequencing (NGS) using the Illumina MiSeq platform. Due to low-fidelity 5′ phosphorylation by the Illumina preparation kit, a single round of PCR was performed using phosphorothioated primers. The amplicons were then purified using AMPure XP beads (Beckman Coulter: Mumbai, India, Cat#A63881) according to the manufacturer protocol and were quantified via FluoroSkan. Sequencing was performed in multiplexed libraries using the NEBNext Ultra II FS DNA Library Preparation Kit and protocol (New England Bio Labs: Ipswich, MA, USA, Cat#E7805L). In brief, the amplicon DNA was fragmented, blunted, and end-repaired with 5′ phosphorylation and 3′ dA-tailing. Hairpin loop adapters were ligated to the fragments and USER enzyme was used for U excision to free the adapter ends. The sample was enriched for adaptor ligated fragments of 150–250 bp via size exclusion AMPure XP bead purification. The libraries were enriched by PCR and barcoded to allow for multiplexed libraries. The libraries underwent a final PCR cleanup with AMPure XP bead purification and were assessed for quality using TapeStation. Pooled libraries were run on the Illumina MiSeq using 300 cycle MiSeq Reagent Nano Kit v2 (Illumina: San Diego, CA, USA, Cat#MS-103-1001).

### 4.5. Sequence Analysis

The resulting Illumina sequences were assembled and aligned using the MAFFT alignment program via online server [72]. The Proviral Sequence Annotation & Intactness Test (ProSeq-IT) tool (https://psd.cancer.gov (accessed on 10 May 2024)) was used to annotate defects throughout the provirus and to infer genetic intactness [39]. The tool infers intactness based on 19 parameters, including sequence length, major splice donor (MSD) site, packaging signal, *gag* start codon status, rev response element (RRE) region status, insertions, deletions, and premature stop codons in *gag*, *pol*, and *env*. The fraction of intact proviruses and those with major internal deletions or mutations in each donor was determined. Proviral populations were also analyzed based on the types and abundance of defects (Figure 1). The fraction of proviruses with defects or deletions in *tat* and *rev* were also determined. To infer “functional” Rev, a multiparameter approach similar to ProSeq-IT was used. Some considerations included the 18Q/60R alpha-helix stability core sites, the M10 dominant-negative mutant (L78D/E79R), the presence of premature stop codons or frameshifts, and deletions within the first 58 amino acids of exon 2 or a deletion of the first 1–10 amino acids in exon 1 [53,54,73]. Due to the multiple subtypes present in the cohorts examined, subtype B/D typically exhibit the 21 bp deletion in the C-terminus (‘TQQSPGT’ motif) relative to other HIV-1 subtypes. Further, it is common to observe the transition mutation of C → T (CAA → TAA; Glutamine → Stop) in subtype C in the C-terminus. These analyses were also performed on previously published datasets for comparison to our dataset. The previously published datasets were obtained from the HIV Proviral Sequence Database (PSD) [74] and include infants treated <5 days after birth (Early Infant Treatment Study (EIT) [38], children on short-term ART (<2.5 years) (EIT delayed-ART) [38], and adults on ART for a median of 9.26 years [33,34,35]. More details on the previously published datasets are provided in Table 2.

### 4.6. Statistical Analysis

All statistical analyses were performed using R (version 4.3.1). If the analysis compared two disjoint sets (e.g., defective versus intact) and only two cohorts, a Fisher exact test was used. If there were more than two cohorts in the comparison, a chi-square test was used. For a comparison of two groups where a clear increase or decrease was observed, a binomial test was used. Note that the Fisher and chi-squared tests were run as 2-tailed tests, while the binomial test was a 1-tailed test.

## Figures and Tables

**Figure 1 viruses-17-00961-f001:**
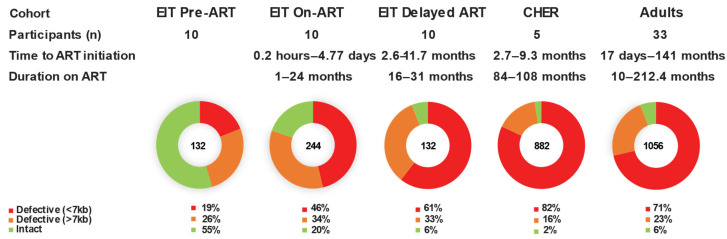
**The fraction of intact and defective proviruses in children and adults.** Red—defective proviruses with large deletions (<7 kb). Orange—defective proviruses with lethal mutations and/or small deletions (>7 kb). Green—intact proviruses.

**Figure 2 viruses-17-00961-f002:**
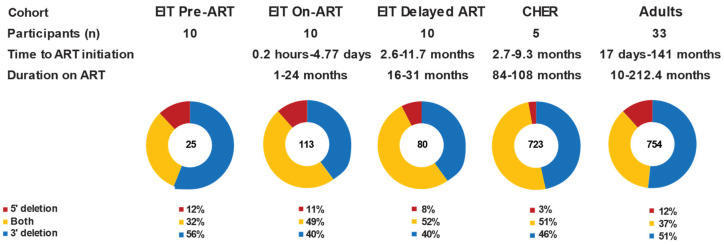
**Proviruses with large internal deletions typically include deletion of the *env* gene.** Blue—proviruses with 3′ large internal deletions. Yellow—proviruses with large internal deletions spanning 5′ and 3′ ends. Red—proviruses with 5′ large internal deletions.

**Figure 3 viruses-17-00961-f003:**
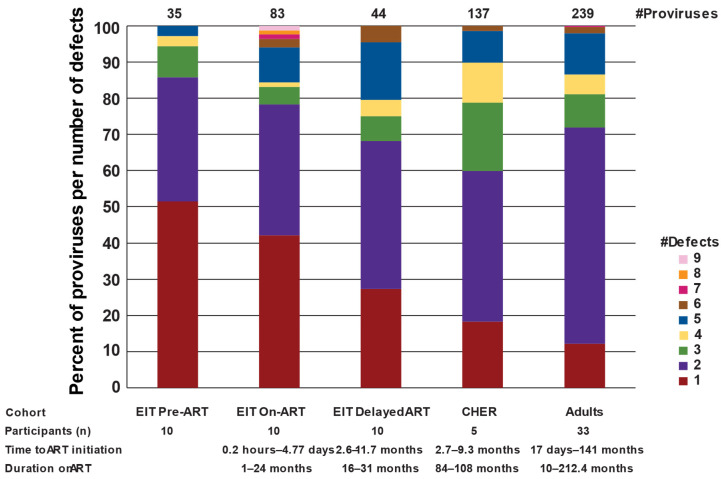
**Number of defects in proviruses >7 kb.** All proviruses that did not have large internal deletions but were not sequence-intact were analyzed for the number of different defects. The different colors indicate the number of proviruses with 1, 2, or more defects. The defects analyzed include MSD defect, packaging signal defects, *gag* start codon defect, insertions (*gag*-*pol*), small deletions (*gag*-*pol*), insertion (*env*), small deletion (*env*), frameshift premature stop codons, non-frameshift premature stop codons, and RRE defect.

**Figure 4 viruses-17-00961-f004:**
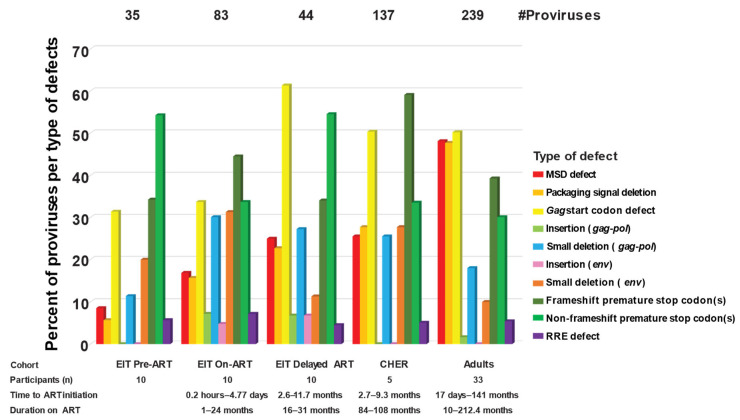
**Different types of defects in proviruses >7 kb.** Red—MSD defect. Light orange—packaging signal defect. Yellow—*gag* start codon defect. Light green—insertions (*gag*-*pol*). Blue—small deletions (*gag*-*pol*). Pink—insertion (*env*) Dark orange—small deletion (*env*). Dark green—frameshift premature stop codons. Green—non-frameshift premature stop codons. Purple—RRE defect.

**Figure 5 viruses-17-00961-f005:**
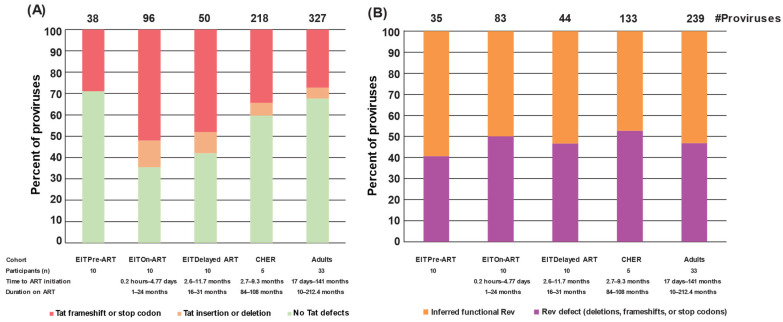
***Tat* and *rev* characterization of *env*-retaining defective proviruses.** (**A**) *Tat* characterization of *env*-retaining defective proviruses (**B**) *Rev* characterization of *env*-retaining defective proviruses.

**Table 1 viruses-17-00961-t001:** Participant demographics and virologic characteristics.

Participant ID	Sex ^A^	Age at ART Initiation (Months)	Time on Suppressive ART (Years)	HIV DNA Copies/10^6^ PBMC on ART	ART Regimen ^B^
ZA-001	F	10.9	8.80	86.3	AZT/3TC/LPV/r
ZA-004	M	2.7	7.92	23.6	AZT/3TC/LPV/r
ZA-006	F	9.0	7.45	46.7	AZT/3TC/LPV/r
ZA-011	F	9.3	7.35	181.5	AZT/3TC/LPV/r
ZA-012	M	5.1	8.60	11.9	AZT/3TC/LPV/r

^A^ Sex: female (F) and male (M). ^B^ ART: zidovudine (AZT), lamuvidine (3TC), ritonavir-boosted lopinavir (LPV/r).

**Table 2 viruses-17-00961-t002:** Number of intact and defective proviruses obtained from donors in the CHER cohort.

Participant ID	Number of NFL Proviral Sequences Obtained ^A^	Number of Sequences		
		Intact ^B^ (%)	>7 kb Defective (%)	<7 kb Defective (%)
ZA-001	139	2 (1.4)	14 (10.1)	123 (88.5)
ZA-004	56	3 (5.4)	21 (37.5)	32 (57.1)
ZA-006	84	2 (2.4)	21 (25.0)	61 (72.6)
ZA-011	450	15 (3.4)	62 (13.7)	373 (82.9)
ZA-012	153	0 (<0.6)	19 (12.4)	134 (87.6)
Total	882	22 (2.5)	137 (15.5)	723 (82.0)

^A^ Near-full-length (NFL). ^B^ Inferred intact defined as proviral genome passing 55 parameters of ProSeq-IT intactness test [39].

**Table 3 viruses-17-00961-t003:** Donor characteristics of previously published studies in HIV persistence in infants, children, and adults.

Study	Age Group	Subtype	# Donors	Median Time of ART Initiation (Range) ^A^	Median Time on Suppressive ART (Range) ^A^
Ho et al. [35]	Adult	B	8	73.5 m (6 m–141 m)	68 m (28 m–108 m)
Bruner et al. [33]	Adult	B	10	3.28 m (17 d–6 m)	106.5 m (10 m–203 m)
Hiener et al. [34]	Adult	B	6	9 m (6 m–12 m)	125.4 m (38.4 m–212.4 m)
Garcia-Broncano et al. [38]	Infant **^B^**	C	10	2.39 d (0.20 h–4.77 d)	12.5 m (1 m–24 m)
	Children **^C^**	C	10	7.15 m (2.6 m–11.7 m)	23.5 m (16 m–31 m)

^A^ Hours (h), days (d), months (m). ^B^ Early Infant Treatment (EIT) Study: EIT pre-ART and EIT on ART. **^C^** EIT control group, delayed ART (EIT Delayed-ART).

**Table 4 viruses-17-00961-t004:** Number of intact and defective proviruses obtained from donors across cohorts.

Participant ID	Number of Proviral Sequences Obtained ^A^	Number of Sequences		
		Intact ^B^ (%)	>7 kb Defective (%)	<7 kb Defective (%)
EIT Pre-ART	132	72 (54.5)	35 (18.0)	25 (26.5)
EIT On-ART	244	48 (19.7)	83 (34.0)	113 (46.3)
EIT Delayed-ART	132	8 (6.1)	44 (33.3)	80 (60.6)
Adults	1056	63 (6.0)	239 (22.6)	754 (71.4)
CHER	882	22 (2.5)	137 (15.5)	723 (82.0)

^A^ EIT Pre-ART, EIT On-ART, EIT Delayed-ART [38], Adults [33,34,35], CHER (this study). ^B^ Inferred intact defined as proviral genome passing 55 parameters of ProSeq-IT intactness test [39].

## Data Availability

All NFL sequences have been deposited in NCBI (PV625462-PV626342) and the NCI Proviral Sequence Database using the PubMed ID of this manuscript. Additional data availability requests can be fulfilled by contacting the corresponding author(s).

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
