# Peer review of "Differential HIV-1 Proviral Defects in Children vs. Adults on Antiretroviral Therapy"

_viruses, 2025, doi:10.3390/v17070961_

Round 1

Reviewer 1 Report

Comments and Suggestions for Authors

Hasson et. al. examine latent HIV in 5 children using state of the art methods. They compare the level of intact, defective and types of defects with other historical provirus data gathered from adults, infants and children. The different cohorts varied in subtype, duration of untreated infection, age, and antiretroviral drugs. They find the 5 children have lower intact intact proviruses than adults. Defective viruses tended to have 3’ defects. They suggest that immune selection pressures may be different among the cohorts.

While it is quite likely that immune pressures are different among the participants in the various cohorts, this data is inadequate for that conclusion. There are numerous other differences in the cohorts that preclude that conclusion, such as the subtype of the virus and duration of infection. Furhermore, there is no information on gender and route of acquisition that are also potentially a contributing factor. As such, the manuscript characterizes the proviral landscape in 5 children. The conclusions with the other cohorts are highly speculative.

It would be ideal to indicate the statistical test used for each comparison. Use of the Fischer exact or Chi square test ignores the differences among the individual participants within a cohort. I am not certain where the “binomial test” was used.

Was the number of PBMC examined if the different cohorts relatively equivalent? Different sampling depth can will also produce provirus landscape differences.

Table 1 appears last. In generally, Table 1 should be the first table.

Zidovudine (AZR).  I have never seen AZR as an abbreviation. Was it supposed to be AZT?

Author Response

Comments and Suggestions for Authors

Hasson et. al. examine latent HIV in 5 children using state of the art methods. They compare the level of intact, defective and types of defects with other historical provirus data gathered from adults, infants and children. The different cohorts varied in subtype, duration of untreated infection, age, and antiretroviral drugs. They find the 5 children have lower intact intact proviruses than adults. Defective viruses tended to have 3’ defects. They suggest that immune selection pressures may be different among the cohorts.

While it is quite likely that immune pressures are different among the participants in the various cohorts, this data is inadequate for that conclusion. There are numerous other differences in the cohorts that preclude that conclusion, such as the subtype of the virus and duration of infection. Furthermore, there is no information on gender and route of acquisition that are also potentially a contributing factor. As such, the manuscript characterizes the proviral landscape in 5 children. The conclusions with the other cohorts are highly speculative.

We agree that there may be many factors that could influence the differences that we observed and that we do not have direct data showing that immune pressure is responsible. We have softened the conclusions in the manuscript to address this concern, and we have highlighted the alternative factors that may influence the differences in the cohorts, such as duration of ART and subtype. We have also indicated that larger studies are needed to determine if the route of transmission or sex differences influence the proviral landscape on long-term ART. Changes are shown in red (lines 36, 265, 285-289, 302-309, 349, and 368).

It would be ideal to indicate the statistical test used for each comparison. Use of the Fischer exact or Chi square test ignores the differences among the individual participants within a cohort. I am not certain where the “binomial test” was used.

We added the specific statistical test used for each comparison (highlight in red in the revised manuscript with each p value).

Was the number of PBMC examined of the different cohorts relatively equivalent? Different sampling depth can will also produce provirus landscape differences.

There were differences in the level of sampling across the cohorts. Table 4 shows the number of proviruses (equivalent to the number of infected cells) that were sampled in each group – 130-244 infected PBMCs in the EIT groups, ~1000 in the adults, and ~900 in the CHER cohort. Although our comparisons were normalized for the numbers of infected cells assayed in each group, we agree that sampling differences is a limitation that we failed to address. We modified the manuscript on lines 207-208, 289, 305-306, and 349 to address this limitation of the study and indicated that larger studies are needed to verify our observations. We also softened our conclusions in the discussion to indicate that the paper drives a hypothesis and encourages larger studies, rather than making any definitive statements.

Table 1 appears last. In generally, Table 1 should be the first table.

Thank you for pointing this out. We have now moved the statement “PBMC were obtained from 5 children…(Table 1)” from the Methods to the first section of the Results so that it appear first in the manuscript.

Zidovudine (AZR).  I have never seen AZR as an abbreviation. Was it supposed to be AZT?

The Reviewer is correct, it has been changed to AZT in Table 1.

Reviewer 2 Report

Comments and Suggestions for Authors

A comparative study analyzed the proportion of defective HIV proviruses in pediatric groups before initiating ART, after short-term and long-term ART, as well as in adults after long-term treatment. The study also evaluated the nature of proviral defects—specifically the localization and size of deletions and the presence of insertions.

The smallest proportion of large deletions was observed in the group with a short duration of ART; as ART duration increased, the proportion of large deletions also increased.

In all groups, the highest proportion of deletions was found at the 3' end of the genome, likely due to template switching during reverse transcription. A significant proportion of solo LTRs—resulting from homologous recombination during the integration stage—was noted in those receiving long-term ART.

In 50% of patients who retained the env gene across all groups, the rev coding region was functionally intact. Tat gene defects occurred more frequently with increasing ART duration; however, tat and rev gene products could still be produced and theoretically targeted by T cells or monoclonal antibodies.

Gag gene defects were detected at a high frequency in all groups. As for defects in the major splice donor site (MSD) and the packaging signal Psi, these were less common than gag defects and occurred less frequently in children than in adults.

The most common defect was the presence of stop codons, most often resulting from APOBEC3G hypermutation. These appear shortly after ART initiation due to the natural selection against intact proviruses.

A general observation relates to the evolution of HIV under ART pressure, indicating an increasing share of defective proviruses due to selection favoring transcriptionally inactive viruses with a maximal number of defects.

This is an extremely interesting article, containing many new findings and making a significant contribution to the understanding of HIV reservoirs, the landscape of defective proviruses, and providing a foundation for developing new approaches to reservoir measurement and potential cure strategies.

The article is ready for publication. It may be advisable—but not necessary—to include an illustration indicating the genomic regions of HIV described in the text, at the authors' discretion.

Author Response

Comments and Suggestions for Authors

A comparative study analyzed the proportion of defective HIV proviruses in pediatric groups before initiating ART, after short-term and long-term ART, as well as in adults after long-term treatment. The study also evaluated the nature of proviral defects—specifically the localization and size of deletions and the presence of insertions.

The smallest proportion of large deletions was observed in the group with a short duration of ART; as ART duration increased, the proportion of large deletions also increased.

In all groups, the highest proportion of deletions was found at the 3' end of the genome, likely due to template switching during reverse transcription. A significant proportion of solo LTRs—resulting from homologous recombination during the integration stage—was noted in those receiving long-term ART.

In 50% of patients who retained the env gene across all groups, the rev coding region was functionally intact. Tat gene defects occurred more frequently with increasing ART duration; however, tat and rev gene products could still be produced and theoretically targeted by T cells or monoclonal antibodies.

Gag gene defects were detected at a high frequency in all groups. As for defects in the major splice donor site (MSD) and the packaging signal Psi, these were less common than gag defects and occurred less frequently in children than in adults.

The most common defect was the presence of stop codons, most often resulting from APOBEC3G hypermutation. These appear shortly after ART initiation due to the natural selection against intact proviruses.

A general observation relates to the evolution of HIV under ART pressure, indicating an increasing share of defective proviruses due to selection favoring transcriptionally inactive viruses with a maximal number of defects.

This is an extremely interesting article, containing many new findings and making a significant contribution to the understanding of HIV reservoirs, the landscape of defective proviruses, and providing a foundation for developing new approaches to reservoir measurement and potential cure strategies.

The article is ready for publication. It may be advisable—but not necessary—to include an illustration indicating the genomic regions of HIV described in the text, at the authors' discretion.

We would like to thank the Reviewer for enthusiasm and strong interest in this work. We apologize if it was not available, but Supplemental Figure 1 illustrates the genomic regions of HIV described in the text.

Round 2

Reviewer 1 Report

Comments and Suggestions for Authors

The authors have addressed the previous concerns. I have no other issues